# Enhancing Long-Term Action Quality Assessment: A Dual-Modality Dataset and Causal Cross-Modal Framework for Trampoline Gymnastics

**DOI:** 10.3390/s25185824

**Published:** 2025-09-18

**Authors:** Fengyan Lin, Jiahao Huang, Zhide Chen, Kexin Zhu, Chen Feng

**Affiliations:** 1College of Computer and Cyberspace Security, Fujian Normal University, Fuzhou 350007, China; qsz20241935@student.fjnu.edu.cn (F.L.); qsz20241923@student.fjnu.edu.cn (J.H.); 2Department of Computer Science, National Sun Yat-sen University, Kaohsiung 518107, China; m073040090@student.nsysu.edu.tw; 3Department of Information Engineering, Fuzhou Polytechnic, Fuzhou 350108, China; fc@fvti.edu.cn

**Keywords:** action quality assessment, causal modeling, temporal feature enhancement, automatic evaluation, human movement

## Abstract

Action quality assessment (AQA) plays a pivotal role in intelligent sports analysis, aiding athlete training and refereeing decisions. However, existing datasets and methods are limited to short-term actions, lacking comprehensive spatiotemporal modeling for complex, long-duration sequences like those in trampoline gymnastics. To bridge this gap, we introduce Trampoline-AQA, a novel dataset comprising 206 video clips from major competitions (2018–2024), featuring dual-modality (RGB and optical flow) data and rich annotations. Leveraging this dataset, we propose a framework comprising a Temporal Feature Enhancer (TFE) and a forward-looking causal cross-modal attention (FCCA) module, which improves action quality assessment by delivering more accurate and robust scoring for long-duration, high-speed routines, particularly under motion ambiguities. Our approach achieves a Spearman correlation of 0.938 on Trampoline-AQA and 0.882 on UNLV-Dive, demonstrating superior performance and generalization capability.

## 1. Introduction

Action quality assessment (AQA) has emerged as a pivotal area of intelligent sports analysis, aiding athlete training, performance evaluation, and refereeing decisions [1,2]. It supports various applications, including skill evaluation [3,4], medical care [5,6], and sports analysis [7,8,9,10,11,12,13,14,15]. In sports, precise AQA is essential for self-assessment, athlete selection, and objective judging. While recent AQA research excels in short, isolated actions, many sports involve long-duration sequences with continuous maneuvers, demanding robust modeling of spatiotemporal dynamics and long-term dependencies. Trampoline gymnastics serves as a prime example, where athletes perform routines of successive aerial 8–12 elements, within 25–35 s sequences [16], highlighting the need for advanced AQA to capture transitions, rhythm, and overall coherence.

Existing research has developed datasets and methods primarily focused on short-term actions in sports such as diving, figure skating, and gymnastics [7,8,9,10,15]. Datasets like AQA-7 [7] and MTL-AQA [8] provide annotations for scores and actions, but they average under 5 s per clip, limiting spatiotemporal modeling for sequences exceeding 20 s. At the same time, methods employ architectures such as C3D [10] and I3D [17] for spatiotemporal feature extraction and score regression [18,19,20,21,22]. However, they are designed for brief sequences, which limits their applicability to prolonged routines. These approaches reveal critical limitations, as short clip lengths hinder comprehensive analysis of long-term dependencies, while models struggle with fine-grained semantics and causal relationships across actions, leading to poor interpretability and inaccuracies in identifying key factors like sequence continuity.

Prior works have attempted to address these issues through advancements in multimodal integration and causal modeling. For instance, some fuse video with skeletal or audio data [11,23], while others incorporate text-assisted frameworks for interpretable evaluations [24,25]. Causal approaches, such as those using cross-modal attention in action recognition [26,27,28], aim to capture cause-and-effect relationships in sequences. Transformer-based models [29,30] focus on sequence dependencies.

Despite these efforts, existing AQA methods typically face two significant challenges [7,18,19,20,21,22,31]. First, existing methods struggle with insufficient feature representation for nonlinear trajectories in high-dynamic sports. Most approaches directly extract and aggregate spatiotemporal features (e.g., via global average pooling) from feature extractors without exploring fine-grained temporal relationships, which neglects distinct action stages (e.g., takeoff vs. landing phases) and fails to identify scoring-critical moments, resulting in blurred representations of complex action. Second, there is an inadequate fusion of multimodal data and a lack of explicit causal modeling for action continuity in long sequences. Single-modality approaches falter in motion-blurred scenarios, and even multimodal ones often ignore inter-modal causal dependencies, limiting effectiveness in domains like trampolining.

Motivated by these challenges, we introduce Trampoline-AQA to complement the research on long-duration motion, a dual-modality dataset comprising 206 clips, averaging 30.05 s in length, from competitions held between 2018 and 2024. The dataset comprises RGB and optical flow data, along with annotations for scores, difficulty levels, and action counts, as detailed in Table 1. In addition, we propose a novel dual-stream framework that fuses RGB video and optical flow modalities, as illustrated in Figure 1. The pipeline segments input videos into clips, extracts optical flow using RAFT [32], and captures spatiotemporal features via backbones such as I3D or X3D (Expanded 3D) [33]. Building on this, we introduce the Temporal Feature Enhancer (TFE) to expand local temporal receptive fields through 1D convolutions, refining representations for complex sequences, and the forward-looking causal cross-modal attention (FCCA) module, which employs history and partial future masks for keyframe-focused causal modeling and fusion of multimodal-feature fusion, thereby uniquely enabling forward-looking causal modeling.

Our contributions fill key research and achieve state-of-the-art Spearman correlations of 0.938 on Trampoline-AQA and 0.882 on UNLV-Dive, outperforming baselines by 5% and validating the framework’s effectiveness and generalization in long-duration, continuous action assessment.

The contributions of this article are summarized as follows:We introduce Trampoline-AQA, the first-ever dual-modality dataset for trampoline gymnastics, which addresses gaps in existing AQA datasets by providing more extended sequences and multimodal data, enabling comprehensive analysis of continuous, complex actions.We developed the Temporal Feature Enhancer (TFE), a novel module that refines spatiotemporal representations from backbones like I3D and X3D by expanding local temporal receptive fields through 1D convolutions.We propose the forward-looking causal cross-modal attention (FCCA) module, which innovatively integrates history masks for past dependencies and partial future masks for controlled forward-looking inference on keyframes, significantly enhancing precision in long-term AQA.

## 2. Related Work

### 2.1. Sport Video Dataset

Sports video datasets serve as essential resources at the intersection of computer vision and sports analytics, where accurate action understanding relies on finely annotated high-quality datasets [34], particularly for high-dynamic sports like trampolining that involve long-duration sequences.

In early studies, datasets such as Olympic Sports [35], Sports-1M [36], and UCF101 [37] provided broad action taxonomies that spanned various sports, including basketball, football, and swimming, but their annotations were limited to coarse category labels. Subsequently, large-scale classification datasets, such as Kinetics-700 [38] and Something-Something V2 [39], introduced finer temporal granularity; however, these prioritize classification over quality regression, and they still lack detailed quality scores or multi-stage action breakdowns, limiting transfer to AQA.

For AQA, specialized datasets include AQA-7 [7] for multi-sport actions, MTL-AQA [8] and FineDiving [9] for diving, FSD-10 [14] for figure skating, FineGym [40] for gymnastics, and LOGO [15] for group dynamics in multi-person videos. Despite providing richer annotations, these datasets [7,8,9,14,40] predominantly focus on short-term actions (e.g., dives averaging 4 s), which limits their applicability to long-duration, continuous movements, such as trampoline routines(averaging 30.05 s).

Existing datasets fail to model high-dynamic, multi-stage sports adequately, due to their short durations and limited multimodal data. To address these gaps, we introduce the Trampoline-AQA dataset, as detailed in Table 1, featuring longer average durations, multimodal data (including both RGB and optical flow), and comprehensive annotations for score, difficulty, and action count. The release of Trampoline-AQA thus establishes a new baseline for spatiotemporal modeling and evaluation of trampoline performances.

### 2.2. Video-Based AQA Methods

Action quality assessment (AQA), at the intersection of computer vision and sports science, was first introduced by Pirsiavash et al. [13] in 2014. They used pose keypoints and SVR for score prediction, which was effective for simple golf swings but proved insufficient for fine-grained evaluation of complex trampoline routines.

As video processing advanced, Parmar and Morris [10] pioneered the use of C3D-SVR and C3D-LSTM frameworks for AQA. To address the inherent subjectivity of the judging, Tang et al. [18] introduced uncertainty-aware score distribution learning (USDL), and Zhang et al. [19] used distributed autoencoders (DAEs) to model score distributions for robustness. However, these approaches still face challenges in fully decoupling spatiotemporal features, particularly in complex scenarios such as trampolining.

Recently, researchers have focused on fine-grained decomposition of action sequences to address performance bottlenecks in AQA. For instance, Yu et al. [20] proposed CoRe, a group-aware contrastive regression framework that learns scoring rules through paired video comparisons. Zhou et al. [21] introduced HGCN, a hierarchical convolutional graph network for aggregating temporal features in continuous actions. Ke et al. [22] developed T^2^CR, which integrates direct and contrastive regression paths for improved generalization. Additionally, Zhou et al. [41] enhanced interpretability with fixed sub-grade prototypes and hierarchical evaluation. AQA, using the Transformer framework, has also emerged. Bai et al. [29] utilized the temporal parsing Transformer to model sequence dependencies, while Fang et al. [30] proposed the action parsing Transformer to extract fine-grained, step-wise representations. Xu et al. [42] proposed the FineParser framework to improve the performance of AQA with human-centric foreground action mask annotations. While these methods improve local discrimination and temporal modeling, they often ignore the global causal structure of long-term actions, limiting their effectiveness in scenarios such as trampolines. In the related field of action recognition, Cai et al. [43] proposed a deep historical LSTM network that accumulates historical states to model temporal evolution for classifying actions in videos. While designed for classification tasks, its historical accumulation approach shares similarities with our causal modeling, though direct application to AQA regression is limited due to task differences.

Despite progress, existing methods struggle with causal modeling in long-duration actions, limiting their performance in dynamic sports like trampolining, where spatiotemporal decoupling is crucial. To address these challenges, approaches that incorporate temporal feature enhancement and causal attention mechanisms can improve the modeling of long-term dependencies and action stages.

### 2.3. Multimodal and Causal-Based AQA Methods

Building on the limitations of video-based AQA methods discussed previously, single-modality methods falter in dynamic scenes (e.g., Spearman correlations dropping below 0.8 due to motion blur in RGB streams). To address these challenges, current AQA research [44] increasingly emphasizes multimodal approaches, integrating data from multiple sources like visual, optical flow, and text modalities to enhance comprehensiveness and accuracy in performance evaluation.

Multimodal AQA has advanced rapidly by integrating diverse sources for assessments. For instance, Zahan et al. [11] fused video and skeletal data for vault quality, while Xia et al. [23] combined video and audio for figure skating. More recently, text-assisted approaches have enabled precise and interpretable evaluations. Zhang et al. [24] proposed a prompt-guided multimodal interaction framework that converts score regression into video-text matching. Moreover, Gedamu et al. [25] proposed unsupervised modeling of fine-grained sub-actions through self-supervised visual–semantic alignment and temporal parsing networks. Zeng et al. [45] proposed a progressive adaptive multimodal fusion network, extracting features from RGB, optical flow, and audio via separate branches with progressive fusion.

At the same time, cross-modal causal modeling is rapidly evolving in the field of video understanding. Patel et al. [26] addressed long-term action recognition in sports videos by modeling cross-modal causal relationships between videos and labeled text. Liu et al. [27] proposed a cross-modal causal relational reasoning framework tailored for event-level visual question answering, which effectively captures the detailed interactive dynamics between visual content and linguistic information. Furthermore, Li et al. [28] introduced causal temporal attention to infer cause-and-effect relationships in action sequences based on video and mask features, enhancing prediction accuracy for diving. Motivated by these studies, multimodal frameworks that fuse optical flow with RGB streams and leverage causal cross-modal attention can overcome the limitations of single-modality approaches.

In summary, existing multimodal methods have advanced standardized sports analysis [46]. However, they still encounter key limitations in handling complex flips and body rotations in long-duration, high-speed scenarios. First, they suffer from insufficient feature representation. Current spatiotemporal models often fail to capture nonlinear changes in trajectory. Second, they provide inadequate multimodal feature fusion. Many approaches fail to leverage complementary information, resulting in suboptimal accuracy gains. Third, the absence of causal modeling restricts insights into action continuity and quality. These gaps motivate the use of multimodal data from datasets like Trampoline-AQA, along with modules for temporal enhancement and causal modeling, to improve representation, fusion, and causality in AQA for trampoline gymnastics.

## 3. Proposed Trampoline-AQA Dataset

In this section, we present the Trampoline-AQA dataset. We provide a detailed description of the dataset construction process, the label annotation strategy, the data statistics, and the data modalities.

### 3.1. Dataset Collection

Since the 2016 Olympics, the scoring rules for trampoline competitions have undergone significant updates. In addition to the traditional difficulty score, execution score, and penalty score, the displacement score and flight time are now new evaluation dimensions [47]. To adapt to this change, we selected hundreds of high-quality videos from authoritative events, including the Olympics, Asian Games, and World Championships, from 2018 to 2024. These videos cover both men’s and women’s individual trampoline events. All videos strictly adhere to the latest competition rules, ensuring the standardization of movements and the diversity of athlete performances. This provides a solid data foundation for subsequent research. Figure 2 illustrates some of the video scenes from the Trampoline-AQA dataset.

The dataset consists of 206 videos that cover complete action sequences with a range of scores and difficulty levels, including complex skills such as front flips, back flips, and twisting combinations. All videos were recorded at a resolution of 1920 × 1080 pixels and 60 fps, allowing for detailed capture of high-speed aerial maneuvers. The raw data was de-duplicated to retain only unique and complete cycles from take-off to landing. Furthermore, all judges’ scoring records were manually verified to ensure consistency between the provided annotations and the actual competition scores.

### 3.2. Label Annotation

We adopted a multi-level annotation strategy to ensure comprehensive and precise action quality assessment. The annotated tags used in this study are listed in Table 2.

At the basic level, each video is labeled with information such as competition type, competition stage, and competition ranking. The scoring annotations include difficulty, execution, flight time, horizontal displacement, penalty score, and total score. To further enhance the granularity of action evaluation and improve upon previous studies [20,22], we analyze the number of actions performed by athletes in each video and assign detailed classification labels to each action video. Additional labels, including athlete information and video duration, are also incorporated.

Regarding the annotation process, we employed a multi-expert collaborative approach to ensure both quality and consistency. One expert initially completed each annotation and subsequently reviewed and refined it by another. This double-review process helps us achieve a high standard of annotation reliability, including manual cross-verification of judges’ scores against officially published scores from the International Gymnastics Federation (FIG) and other authoritative sources. Specifically, two independent experts compared the scoring information appearing in the video clips with the official scoring sheet and resolved any discrepancies by consensus. However, inter-rater reliability statistics are not available, as the dataset only includes final aggregated scores; FIG or other federations do not publicly release individual judges’ raw scores.

Subjective elements in human scoring (e.g., execution scores) are inherently normalized across competitions and judges through the standardized international trampoline rules, which minimize variability by providing clear guidelines for evaluation. All ground-truth scores in the Trampoline-AQA dataset are official final scores (ranging from 0 to 65) published by FIG or the Chinese Gymnastics Association on their official results portals. These scores have already undergone rigorous normalization processes, including the discarding of the highest and lowest execution scores, reviews by senior judging panels for difficulty and time-of-flight deductions, and video replay reviews in cases of significant discrepancies. Therefore, no additional normalization was applied during dataset construction, as the scores are authoritative and directly adopted for our model’s experiments.

### 3.3. Dataset Statistics

The Trampoline-AQA dataset (Code and dataset: Trampoline-AQA, accessed on 20 July 2025) consists of 206 videos, totaling 3.06 h in duration. The average duration of each action sequence is 30.05 s, with 114 male samples and 92 female samples, which aligns with the characteristics of short-duration, highly dynamic trampoline gymnastics.

To ensure rigorous model training and evaluation, we divided the dataset into two parts—a training set comprising 75% of the data (155 videos) for model development and parameter optimization, and a test set comprising 25% of the data (51 videos) for the final evaluation of model generalization.

Figure 3 presents the distribution of the final scores for both the training and test sets. The x-axis represents the final score, while the y-axis indicates the number of samples for each score. In the training set, scores are primarily concentrated between 50 and 60 points, with a mean of 47.89 (red dashed line) and a median of 53.22 (green dashed line). The test set shows a similar distribution, though it contains fewer high-scoring samples, and the scores are more dispersed. The mean score in the test set is 50.09, and the median is 53.52, again illustrating a higher median compared with the mean. Overall, the score distributions in the training and test sets are consistent, although the test set is relatively sparser. To further quantify the consistency of ground-truth scores under our random 75%/25% split, we conducted a Kruskal–Wallis test between the training and test sets, yielding H = 0.026 and *p* = 0.87 (Cohen’s d = 0.156). These results indicate no statistically or practically significant differences, ensuring that the model is trained and evaluated on statistically indistinguishable distributions, thereby avoiding any exploitation of judge-specific biases.

### 3.4. Dataset Modality

The Trampoline-AQA dataset offers bimodal visual data, consisting of both video streams and optical flow, to support a comprehensive motion analysis. The video stream modality provides consecutive frames that capture athletes’ visual appearance and movement patterns during competition, forming the primary visual reference for quality assessment. In contrast, the optical flow modality is derived from inter-frame pixel displacement calculations, capturing dynamic motion characteristics through velocity vectors and directional changes inherent in trampoline gymnastics.

Figure 4 presents the motion analysis results of the bimodal video stream and optical flow data at different performance stages in the Trampoline-AQA dataset, including the “rising phase”, “performance phase”, and “drop phase.” The first two rows illustrate normal performances, while the last row demonstrates abnormal cases, such as color distortion and blur caused by high-speed motion.

In regular performances, the video stream shows natural and clear colors, and the optical-flow feature maps accurately capture the athlete’s motion trajectories and dynamics. During the “performance phase”, the motion trajectories are concentrated and smooth, reflecting both the continuity and precision of the movements. The “rising phase” and “drop phase” display evenly distributed motion information, clearly indicating the initiation and termination of actions.

In contrast, in abnormal performances (as shown in the last row), rapid directional changes and high-speed maneuvers, especially during the “drop phase”, result in substantial color distortion and overexposure in the video stream. These issues significantly reduce the quality of visual information. They also cause a considerable loss of motion information in the optical-flow maps, with discontinuous vector distributions and even blank areas.

## 4. Methods

To address the AQA task, we propose a dual-stream framework, as illustrated in Figure 5. The method processes input videos by dividing them into multiple clips and extracting spatiotemporal features from both video streams and optical flow data. For the optical flow, the feature maps are extracted using RAFT [32]. An optical flow anomaly detection module (OFA) then filters out maps that lack significant motion, ensuring the quality of the dynamic information.

In parallel, video stream clips are processed by I3D to extract spatial pose and appearance features, while X3D is used on optical flow clips to capture dynamic temporal characteristics. To further enhance feature representation, the Temporal Feature Enhancer (TFE) module strengthens local action features. The forward-looking causal cross-modal attention (FCCA) module introduces spatiotemporal causal modeling by leveraging historical and future masks, thereby improving keyframe discrimination and cross-modal feature fusion, which integrates optical flow and video stream features. The scoring prediction module then produces quantitative assessments of action quality.

### 4.1. Optical Flow Sequence Extraction and Sampling

For each video input, an optical flow sequence will be output after extraction and sampling. The complete workflow of optical-flow extraction and sampling is shown in Figure 6. To extract optical flow, we first preprocess the video clips. The preprocessing steps include downsampling the video to 25 fps, clipping, and aligning the segments to ensure the integrity of each action sequence. We also apply image enhancement techniques such as illumination and contrast adjustment to improve frame quality.

Next, we use the pre-trained RAFT [32] model, trained on the FlyingThings3D optical flow dataset, to calculate the optical flow for the video frame sequence $V={\left\{v_{i}\right\}}_{i=1∼N}$. The resulting optical flow sequence is $F={\left\{f_{i}\right\}}_{i=1∼N-1}$, where N is the number of video frames.

Although RAFT is effective at capturing the optical-flow features of athletes, its process of extracting optical flow is susceptible to image noise. To address this, we introduce the composite Optical Flow Anomaly detection (OFA) module, which employs a triple verification mechanism that involves color distribution analysis, optical flow intensity assessment, and temporal continuity validation. This ensures the precise identification of abnormal optical-flow feature maps.

When high saturation artifacts occur on the optical-flow feature map, the pixel distribution deviates from normal motion patterns. To detect such anomalies, we define the saturation anomaly detection function:(1)$$A_{c}=\frac{1}{W\times H}\sum_{x=0}^{W-1}\sum_{y=0}^{H-1}L\left(S\left(x,y\right)>q\right)$$

In Equation (Equation 1), L(·) is an indicator function, S(x,y) represents the saturation value of the pixel at (x,y), *q* is an empirical threshold, and *W* and *H* are the width and height of the image. When $A_{c}$ exceeds a preset critical value $\tau_{s}$, the current optical-flow map is determined to have a color distortion.

Optical-flow feature maps may also experience situations where motion information is missing. To identify such cases, we construct a motion intensity mapping based on the Euclidean distance of the optical flow field and establish a motion intensity threshold equation:(2)$$A_{m}=\frac{1}{W\times H}\sum_{x,y}I\left(|\vec{V}{\left(x,y\right)|}_{2}>k\right)$$

Here, I(·) is an indicator function, $\vec{V}\left(x,y\right)$ represents the motion vector at pixel (x,y), and *k* is a dynamically adjusted intensity threshold based on the action type. If the overall motion intensity $A_{m}$ falls below the threshold $\tau_{m}$, it indicates the presence of missing motion information.

A feature map is retained only if it satisfies both $A_{c}\leq \tau_{s}$ and $A_{m}\geq \tau_{m}$. Feature maps failing either condition are discarded, ensuring that only high-quality data are preserved for robust motion analysis.

Athletes may occasionally exhibit uncoordinated movements or errors, resulting in oversaturated optical-flow colors or missing motion information. To further address these anomalies and prevent misjudgments, we developed a weighted temporal difference algorithm:(3)$$\bar{D}=\frac{1}{2i}\sum_{k=1}^{i}\left[D\left(F_{t},F_{t-k}\right)+D\left(F_{t},F_{t+k}\right)\right]$$

Here, $\bar{D}$ represents the mean temporal difference between the current frame $F_{t}$ and the sequence of frames $F_{i}={\left\{f_{i}\right\}}_{i=t-k∼t+k}$ formed by *i* frames before and after, and $D(F_{a},F_{b})$ denotes the pixel-wise difference between two frames. If $\bar{D}$ exceeds the threshold $\tau_{d}$, the current frame is considered to contain action incoordination or error, and the optical-flow map is retained.

Through the OFA module, unsuitable optical-flow feature maps are removed from the sequence *F*. We then extract samples $V^{\prime }={\left\{v_{i}\right\}}_{i=1∼M}$ and $F^{\prime }={\left\{f_{i}\right\}}_{i=1∼M}$ at fixed intervals ⌊N/M⌋ from the video frame sequence *V* and the optical flow frame sequence *F*, where *M* denotes the number of sampled frames, and *N* represents the total number of frames. This sampling strategy reduces data redundancy and ensures reliable motion features for subsequent action quality assessment.

### 4.2. Feature Extraction

To extract spatiotemporal features from video data and optical flow effectively, we utilize two backbone models, I3D and X3D, which process video clips and optical flow clips, respectively. The selection of I3D for RGB video streams and X3D for optical flow is based on established literature. I3D is chosen for its effectiveness in capturing spatial and temporal information from appearance-based data [17], while X3D is selected for motion-centric optical flow due to its efficient architecture and performance on motion-intensive tasks [33], as supported by two-stream frameworks [11,48]. This choice ensures complementary feature extraction, with criteria centered on pre-training datasets and empirical advantages in handling respective modalities.

We utilize a pre-trained I3D model on the Kinetics-400 dataset to perform spatiotemporal feature extraction on sampled video sequences. The final classification layer of the I3D model is removed to obtain high-dimensional feature embeddings from the penultimate layer. The sampled video sequences $V^{\prime }={\left\{v_{i}\right\}}_{i=1∼M}$ are grouped into clips of every 16 frames. Each video clip is processed through the I3D model for feature extraction, resulting in a feature vector that encapsulates both spatial and temporal information. These feature vectors are subsequently aggregated for downstream analysis. The output features from I3D have dimensions (T,C=1024), where *T* denotes the temporal length determined by the sampling process, and *C* represents the channel dimension, indicating the number of feature channels extracted at each temporal location.

For feature extraction of the optical flow sequences $F^{\prime }={\left\{f_{i}\right\}}_{i=1∼M}$, we use an X3D model pre-trained on the Kinetics-600 dataset with a configuration similar to I3D. The final classification layer is removed to extract intermediate feature representations. The optical flow sequences are similarly divided into multiple clips, with each flow clip input into the X3D model to obtain motion-centric feature embeddings. These features complement the spatial and temporal information extracted from the RGB video clips. The initial output features from X3D have dimensions (T,C=2048). To ensure dimensional alignment with I3D features, we apply a linear projection layer to reduce the channel dimension to 1024, resulting in aligned features of (T,C=1024).

Since both modalities are sampled from the same number of frames *M* at fixed intervals, the temporal dimension *T* is inherently aligned between RGB and optical flow sequences, eliminating temporal misalignment risks.

The features extracted from I3D and X3D are aligned in dimensions to ensure consistency across different modalities. The resulting feature vectors serve as inputs for action quality assessment.

### 4.3. Temporal Feature Enhancement

I3D and X3D are deep learning models designed for video understanding tasks, offering strong spatiotemporal feature learning capabilities. However, traditional backbone feature extractors often exhibit insufficient performance when generating spatiotemporal features in high-speed motion scenarios. To address this limitation, inspired by Huang et al. [49], we propose a novel module called the Temporal Feature Enhancer (TFE), a convolutional local semantic-enhancement module designed to strengthen the temporal-feature representations extracted by networks such as I3D and X3D. Unlike Huang et al. [49], who proposed cascaded 1D convolutions with a bottleneck structure for semantic integration and temporal uniformity, TFE introduces sequence-adaptive kernels and attention weights for dynamic, keyframe-focused enhancement. The architecture of the Temporal Feature Enhancer is illustrated in Figure 7.

Within the TFE module, the spatiotemporal features of the video stream ($F^{r}$) and optical flow ($F^{f}$) extracted by I3D or X3D are combined as the input sequence features $F_{1}$. Then, the global information from $F_{1}$ is extracted through a global pooling operation, generating the global feature representation $F_{1}^{\prime }$. This global representation is then processed by a two-layer fully connected network to generate sequence-adaptive convolutional kernels for dynamic feature adjustment.

The input features $F_{1}$ are then padded and subjected to pixel-wise convolution operations. Finally, local dynamic features $F_{2}$ are extracted using a fully connected layer and ReLU activation, effectively capturing local dynamic patterns within the sequence.

Meanwhile, the module calculates attention weights for each time step using a fully connected layer followed by a sigmoid activation. These weights are applied to $F_{2}$ by weighted fusion with $F_{1}$, highlighting the contribution of keyframes. Finally, a residual connection adds the original features $F_{1}$ to the enhanced features $F_{2}$, producing $F_{E}$. This approach preserves the original information while substantially improving the network’s ability to model dynamic patterns.

For the spatiotemporal sequence features $F_{1}$ of the video, where *T* represents the temporal length of the video sequence, the enhanced features are denoted as $F_{E}$, and the adaptive convolution kernel is denoted as *K*, with *S* representing the size of the kernel. Here, FC denotes the fully connected layer. The operation of the Temporal Feature Enhancement module can be defined as follows:(4)$$\begin{array}{cc} F_{1}^{\prime } & =\mathrm{GlobalPool}\left(F_{1}\right)\in R^{1\times 1024},\\ K & =\mathrm{FC}\left(F_{1}^{\prime }\right)\in R^{S},\\ F_{2} & =\mathrm{FC}\left(\mathrm{RELU}\left(\mathrm{FC}\left(\mathrm{Conv}1D\left(F_{1},K\right)\right)\right)\right)\in R^{T\times 1024},\\ F_{E} & =F_{1}+\mathrm{Sigmoid}\left(\mathrm{FC}\left(F_{1}\right)\right)⊙F_{2}\in R^{T\times 1024}. \end{array}$$

### 4.4. FCCA Module

We propose the FCCA module, whose primary objective is to enhance discriminative feature representation in action sequences through both spatiotemporal causal modeling and cross-modal feature fusion. The detailed architecture of the FCCA module is illustrated in Figure 5. The module consists of two main components—forward-looking causal attention (FCA), which uses historical and future masks for spatiotemporal causal modeling, and cross-modal relation-enhanced dual attention (CREDA), which focuses on feature fusion between modalities.

The FCA component is a spatiotemporal causal modeling mechanism that utilizes historical and future masks to adaptively allocate attention weights and capture causal relationships within action sequences. This enables the extraction of deeper latent optical-flow features ($F_{c}^{f}$) and latent video-flow features ($F_{c}^{r}$).

Specifically, the FCA module analyzes the influence of historical frames on the current frame within an action chain and the role of brief future frames in maintaining action continuity. This design, with an emphasis on keyframes, allows for a more accurate understanding of the balance and stability of the current action.

In the spatiotemporal feature map, we introduce the causal masking matrix $M_{\mathrm{future}}$ to constrain attention to only historical and current frames. The causal masking matrix is defined as follows:(5)$$M_{\mathrm{history}}\left(i,j\right)=\left\{\begin{array}{cc} 1, & \mathrm{if}\;i\geq j,\\ 0, & \mathrm{otherwise}. \end{array}\right.$$

Furthermore, a look-ahead mask $M_{\mathrm{future}}$ is used to allow the model to have a brief window into future frames, thus improving the coherence of local actions and their temporal relationships. The lookahead causal masking matrix is defined as follows:(6)$$M_{\mathrm{future}}\left(i,j\right)=\left\{\begin{array}{cc} 1, & \mathrm{if}\;j-i\leq t,\\ 0, & \mathrm{otherwise}. \end{array}\right.$$

By integrating a historical mask and a short future look-ahead mask, FCA explicitly bounds long-range dependencies—the historical mask blocks access to frames beyond the current time step, while the future mask permits only a small, tunable window (t). This design curbs over-reliance on distant future evidence, emphasizes local temporal continuity, and reduces bias from early events. With a brief look-ahead, FCA adaptively allocates attention to quantify how past frames influence the current action and how limited future context stabilizes the assessment, without inducing premature global judgments. This approach enables the model to capture the causal chain of actions more accurately, particularly improving the modeling of action stability in dynamic and complex scenarios. The FCA operation is defined as follows:(7)$$\begin{array}{cc} FCA(Q,K,V) & =\mathrm{Softmax}\left(\frac{QK^{⊤}}{\sqrt{d_{k}}}⊙M\right)V \end{array}$$
where Q,K,V represent the query, key, and value matrices of the input features. *M* is the combination of the historical-causal mask matrix $M_{\mathrm{history}}$ and the look-ahead causal-coding matrix $M_{\mathrm{future}}$, defined as follows:(8)$$M=M_{\mathrm{history}}+M_{\mathrm{future}}$$

We propose the CREDA module to achieve effective information exchange between video streams and optical flows—it queries video features with optical-flow vectors and vice versa, allowing each modality to guide the other’s attention. This bidirectional querying, implemented within the FCCA module, automatically aligns the two streams and yields coherent, fused representations without additional alignment layers.

Specifically, the interaction between video features and optical-flow features is formulated as follows:(9)$$A_{v}=\mathrm{Softmax}\left(\frac{Q_{f}K_{r}^{⊤}}{\sqrt{d_{k}}}\right)V_{r}$$(10)$$A_{f}=\mathrm{Softmax}\left(\frac{Q_{r}K_{f}^{⊤}}{\sqrt{d_{k}}}\right)V_{f}$$

Here, $A_{v}$ is the attended video feature map that incorporates motion information, and $A_{f}$ is the attended optical-flow feature map that incorporates appearance information. $Q_{f}$ and $Q_{r}$ represent the queries for optical-flow and video features, respectively. $K_{r}$ and $K_{f}$ represent the keys for video and optical-flow features, respectively. $V_{r}$ and $V_{f}$ represent the values for video and optical-flow features, respectively.

Through the cross-modal attention mechanism, the CREDA module enables effective information exchange and feature enhancement between modalities, improving the model’s global understanding of action sequences. Finally, global pooling is applied to fuse cross-modal information and generate the global feature representation $F^{g}$. Overall, the FCCA module significantly enhances spatiotemporal feature representation, providing robust support for action quality assessment.

### 4.5. Score Prediction and Loss Function

To better capture the distribution and uncertainty of video quality scores, we introduce a score distribution autoencoder. This autoencoder encodes the input feature *x* into the parameters of a Gaussian distribution, μ(x) and $\sigma^{2}\left(x\right)$, where the mean μ(x) represents the predicted score and the variance $\sigma^{2}\left(x\right)$ quantifies the uncertainty. The probability density function g(y;θ(x)) maps the feature *x* to the random variable *y*:(11)$$g\left(y;\theta \left(x\right)\right)=\frac{1}{\sqrt{2\pi \sigma^{2}\left(x\right)}}\exp\left(-\frac{{\left(y-\mu \left(x\right)\right)}^{2}}{2\sigma^{2}\left(x\right)}\right)$$

The random variable *y* is generated by sampling the noise ε from the standard normal distribution N(0,1) and combining it with the mean and variance:(12)y=μ(x)+ε·σ(x)

To align the distribution of predicted scores with the actual data distribution, we calculate the mean squared error (MSE) between the predicted scores (pred) and the actual scores (true), incorporating the mean constraint μ(x) to improve stability and accuracy. The loss function is defined as follows, where *N* is the batch size:(13)$$L_{\mathrm{MSE}}=\sum_{i=1}^{N}\left[{\left({\mathrm{pred}}_{i}-{\mathrm{true}}_{i}\right)}^{2}+{\left({\mathrm{true}}_{i}-\mu \left(x\right)\right)}^{2}\right]$$

Furthermore, to ensure that the distribution of latent variables approximates the standard normal distribution N(0,1), we introduce the Kullback–Leibler (KL) divergence loss $L_{\mathrm{KL}}$. The KL divergence measures the difference between two probability distributions and is defined as follows:(14)$$L_{\mathrm{KL}}=-0.5\sum_{i=1}^{N}\left(1+\log\left(\sigma_{i}^{2}\right)-\mu_{i}^{2}-\sigma_{i}^{2}\right)$$

Finally, the total loss function $L_{\mathrm{Total}}$ used for constrained training combines the mean square loss and the KL divergence loss, with α as a regulation parameter to balance the two terms:(15)$$L_{\mathrm{Total}}=L_{\mathrm{MSE}}+\alpha \cdot L_{\mathrm{KL}}$$

## 5. Experiment

In this paper, the model was trained on a mainframe computer equipped with dual NVIDIA GeForce RTX 3090 GPUs (each with 24 GB of memory), an Intel Xeon Silver 4310 CPU at 2.10 GHz, and 26 GB of RAM, running Ubuntu 22.04 with PyTorch 1.13.1 and Python 3.8.

### 5.1. Datasets

In this paper, we evaluate our method on two challenging datasets, our Trampoline-AQA dataset and the UNLV-Dive dataset.

The Trampoline-AQA dataset is divided into a training set (155 samples) and a test set (51 samples), with a ratio of 3:1. The division of the training and test sets for Trampoline-AQA is shown in Figure 3.

The UNLV-Dive dataset [10] contains 370 videos from the men’s 10 m platform semi-finals and finals at the 2012 London Olympics. Each video sample includes the total score, difficulty level, and segmentation labels. The difficulty level ranges from 2.7 to 4.1, and the total score ranges from 21.6 to 102.6. In the original split, 300 videos are used as the training set, and the remaining 70 videos are used as the test set.

### 5.2. Evaluation Metrics

Following the experimental setup in existing works [7,18,19,20,21,22,31,50,51,52,53,54,55,56,57,58,59], we use Spearman’s rank correlation coefficient (SRC) as the evaluation metric. The SRC is a classical non-parametric statistical measure that effectively assesses the monotonic relationship between predicted and true scores by comparing their ranking consistency. Its sensitivity to monotonic relationships makes it a standard evaluation metric in AQA research, as it effectively tests whether the prediction model’s assessment of action quality aligns with expert evaluations. The SRC coefficient ρ is calculated as follows:(16)$$\rho =\frac{\sum_{i}^{M}\left(m_{i}-\bar{m}\right)\left(n_{i}-\bar{n}\right)}{\sqrt{\sum_{i}^{M}{\left(m_{i}-\bar{m}\right)}^{2}\sum_{i}^{M}{\left(n_{i}-\bar{n}\right)}^{2}}}$$
where *m* and *n* denote the ranking of two sequences, $\bar{m}$ and $\bar{n}$ are the mean ranking of the corresponding sequence, and *M* is the number of series. The value of ρ ranges from −1 to 1, with higher values indicating better model performance.

### 5.3. Implementation Details

We adopt the Inflated 3D ConvNet (I3D) and Expanded 3D (X3D), both pre-trained on Kinetics, as our video feature extractors. The clip and query sizes for Trampoline-AQA and UNLV-Dive are set to 22 and 6, respectively. These parameters are determined based on the length of the videos and extensive comparison experiments, demonstrating optimal performance.

For both training and testing, we set the batch size to 4 and run the training for 100 iterations. The Adam optimizer is used with a learning rate of $1\times 10^{-4}$. The FCCA module is configured with eight attention heads. The threshold parameters $\tau_{s}$, $\tau_{m}$, and $\tau_{d}$ in the OFA module are set to 0.8, 0.8, and 3, respectively. The sequence of optical-flow feature maps selected by the OFA module has been saved in advance and set as the input of the optical flow sequence. Following the experiment settings in [9,18,20,21,22,28,42], we select 75% of the samples for training and 25% for testing in all experiments. It should be noted that in all experimental settings on the Trampoline-AQA dataset, only the difficulty level labels are used, and no additional manually annotated trampoline classification labels are used. The remaining settings are all original.

### 5.4. Comparison with Baselines

Comparison with state-of-the-art methods: Table 3 and Table 4 present a performance comparison between our method and state-of-the-art approaches on the Trampoline-AQA and UNLV-Dive datasets. The results demonstrate the superiority of the proposed method. To enhance the rigor of our evaluation and address concerns about result reliability, we provide dataset statistics in Section 3.3, including a Kruskal—Wallis test confirming no significant differences in score distributions between training and test sets (*p* = 0.87). This ensures the fairness of our comparisons.

The Trampoline-AQA dataset: Table 3 provides a comparative analysis of the action quality assessment methods published from 2019 to 2025. The SRC value of our proposed method reaches 0.938, significantly outperforming all baseline methods. Our method shows a 2.7% improvement compared with the best baseline, FineCausal (0.911). Compared with USDL (0.759), the improvement is 17.94%. Notably, our method also shows a significant improvement over the CoRe approach, establishing a new performance benchmark on this dataset.

Table 3 shows that the T^2^CR, FineParser, and FineCausal models, which utilize multimodal data, significantly outperform the single-modal baselines (USDL, DAE, and HGCN). This advantage stems from the complementary motion information provided by multimodal data, enabling the models to more comprehensively understand the details of actions in videos and thereby improve evaluation accuracy. Specifically, FineCausal explicitly captures the temporal dependencies in action sequences through causal modeling, achieving the best performance among all baselines.

In the experimental setup, given the characteristics of the dataset, FineParser and FineCausal use grayscale optical-flow maps instead of the manually annotated mask maps used in the original paper. After removing unclear or indistinguishable motion information and noisy optical-flow maps using the OFA filter, the models focus on practical human motion information. This replacement does not affect the validity of the comparative experiments between the two models. Unlike FineCausal’s spatiotemporal causal attention, the forward causal attention module proposed in this paper models action causal chains through historical-future masks, supplemented by TFE (Temporal Feature Enhancer), to enhance spatiotemporal feature extraction capabilities, collectively enabling the model’s performance to surpass that of FineCausal.

To further illustrate the advantages of our method over other methods, we visualize the predictions on the first 20 test samples, as shown in Figure 8. The results are compared with the USDL, CoRe, and T^2^CR methods. Specifically, the line graph in Figure 8a shows the predictions of USDL, CoRe, T^2^CR, our method, and the ground truth labels. The USDL method exhibits substantial deviations from the ground truth on several samples. In contrast, our method matches the ground truth closely, with only minor errors. The reason for the substantial deviations in the USDL method is that it directly uses aggregated and pooled I3D features for uncertainty modeling, while ignoring the temporal nature of these features. In contrast, our method addresses this issue through the TFE module, which enhances local dynamic patterns.

Figure 8b provides detailed prediction examples for two sample actions. For the high-score sample, the true label is 58.52, with CoRe and T^2^CR predicting 57.45 and 57.52, respectively. Our method predicts 58.10, which is closest to the actual value. Similarly, for the low-scoring sample with a proper label of 23.15, our prediction is also closer to the ground truth value. Unlike the dual-channel video stream framework used by T^2^CR, our framework utilizes complementary visual flow and optical flow to capture more motion information. In addition, the score autoencoder we adopt achieves fine-grained score regression by modeling the score distribution.

As reviewed, the superior performance of our model stems from three complementary design choices. First, the Temporal Feature Enhancer (TFE) enlarges the local temporal receptive field, enabling the network to capture fine-grained motion details that are easily blurred by long-range pooling. Second, the forward-looking causal cross-modal attention (FCCA) establishes explicit causal chains between successive sub-actions, enabling the model to reason about how early errors propagate and affect later quality, which is crucial for routines lasting 25–35 s. Third, the score autoencoder fits the entire score distribution rather than regressing a single scalar, yielding more accurate and stable predictions. These components jointly push the SRC to 0.938.

The UNLV-Dive dataset: As shown in Table 4, our method achieves notable improvements in SRC compared to previous work. Our approach achieves comparable performance to these methods, with an SRC of 0.863 versus a benchmark of 0.86 without explicit difficulty labeling and 0.882 versus the benchmark score of 0.88 for methods that include difficulty labels. These results indicate that even without explicitly relying on difficulty-level annotations, our method matches or slightly exceeds the best available results, highlighting the robustness and generalization capabilities of our framework for action quality assessment.

The superior performance of our model can be attributed to the following key factors. First, the Temporal Feature Enhancer effectively improves the model’s understanding of complex action details by capturing motion dependencies over long periods. In addition, the causal modeling module further enhances the model’s understanding of actions by establishing causal relationships between actions. These designs enable our model to perform excellently in different difficulty levels and action scenarios.

### 5.5. Ablation Study

In this section, we perform ablation experiments on the Trampoline-AQA dataset to demonstrate the contributions of the proposed modules and the impact of key hyperparameters.

#### 5.5.1. Optical Flow Ablation Experiment

Table 5 presents the ablation results for the optical flow modality, evaluating the individual and combined effects of the optical flow anomaly detection (OFA) module and the forward-looking causal attention (FCA) mechanism across two 3D convolutional architectures: I3D and X3D. The baseline method used in the ablation experiments in Table 5 and Table 6 is USDL [18], which serves as the foundation for evaluating the contributions of our proposed modules.

For the I3D architecture, the addition of the OFA module increases the SRC value from 0.789 (baseline) to 0.801. Further integration of the FCA mechanism raises the SRC to 0.853, demonstrating its effectiveness in enhancing optical-flow feature modeling and noise suppression. Similarly, in the X3D architecture, the OFA module increases the SRC from 0.806 to 0.821, and the addition of FCA further increases it to 0.904. These results highlight the significant benefits of combining OFA and FCA.

#### 5.5.2. RGB Ablation Experiment

Table 6 shows the ablation results for the video stream modality, comparing the effects of temporal attention (TA) and the FCA mechanism in the I3D and X3D models. Although both TA and FCA are mechanisms based on attention mechanisms, their designs focus on different aspects of the feature space. TA focuses on capturing long-term temporal dependencies, while FCA emphasizes long-term temporal causal modeling capabilities. Using both mechanisms simultaneously may lead to attention pattern redundancy or conflicts, thereby reducing the model’s ability to extract complementary information.

For I3D, introducing TA raises the SRC from 0.759 to 0.822, indicating that TA enhances temporal modeling. Using FCA further increases the SRC to 0.837, a 1.5% improvement over TA alone, confirming FCA’s significant impact on causal reasoning.

For X3D, the addition of the TA mechanism increases the SRC value from 0.816 to 0.862, a 0.046 improvement, demonstrating the method’s good cross-architecture adaptability. The FCA mechanism improves the SRC value to 0.893, a 3.1% improvement over using TA. This indicates that FCA can significantly enhance the video stream’s feature modeling capabilities.

Through experiments on the video stream under different base models, the critical role of temporal modeling and causal attention mechanisms in multimodal feature learning tasks has been validated, and the stability and effectiveness of the proposed method have been demonstrated.

#### 5.5.3. Temporal Feature Enhancer Ablation Experiment

Table 7 presents the impact of Temporal Feature Enhancer (TFE) across different modalities using the I3D backbone. The optical flow stream is preprocessed by the OFA module. The results show that TFE improves SRC by 8.1% for optical flow and by 3.5% for the video stream.

This demonstrates that the TFE effectively enhances feature modeling through 1D convolutions and local attention, enabling the optical flow stream to capture dynamic athlete features more effectively than the video stream. Consequently, after enhancement by the TFE, the optical flow stream exhibits superior temporal feature modeling compared with the video stream.

#### 5.5.4. Further Analysis

Table 8 reports a factorial hyperparameter sweep over the temporal convolution kernel size and the causal attention window. We evaluate k∈{1,3,5,7} and t∈{1,2,3,4} on the Trampoline-AQA test set and measure SRC. The best configuration (*k* = 5, *t* = 3) attains an SRC of 0.938, yielding a 2.4% improvement over the smallest setting (*k* = 1, *t* = 1; 0.914). Performance increases with the temporal receptive field up to *k* = 5 but degrades at *k* = 7, suggesting that excessive temporal context introduces noise for short trampoline routines. Similarly, *t* = 3 offers the best trade-off between causal coverage and computational cost, with larger windows bringing no additional gains.

### 5.6. Visual Comparison of Optical Flow Filtering

To intuitively demonstrate the effectiveness of the OFA module in filtering optical flow sequences, Figure 9 compares input data features under different processing strategies. The figure is organized in three rows—the first row presents the original video frames, the second row displays the optical-flow feature maps processed by the OFA module, and the third row shows the optical-flow feature maps without OFA processing.

The first row (a) presents the original video frames, which provide visual appearance information of the athlete and record the spatiotemporal trajectory of their actions, serving as a baseline for motion analysis. The second row (b) shows the optical-flow feature maps after processing with the OFA module. These maps exhibit clear and consistent motion vectors, with dynamic information effectively extracted and background noise significantly reduced.

In contrast, the third row (c) displays the optical-flow maps without OFA processing. Here, the maps suffer from obvious noise interference, disordered motion vectors, blurred features, and poor extraction of dynamic information. This degradation can weaken the model’s ability to interpret action details, affecting the accuracy of motion analysis.

In summary, the OFA module effectively eliminates noise caused by RGB color distortion and insufficient motion information. It significantly enhances the expressive power of optical-flow feature maps, increasing attention to motion regions and providing subsequent models with higher-quality input features for action analysis.

To assess efficiency for real-time applications, we evaluated OFA’s efficiency on the Trampoline-AQA dataset. It processes an average 31-s video (25 fps) in 45.07 s (0.06 s/frame). Additionally, it reduces $A_{c}$ (Equation (Equation 1)) by 74.6% (from 0.4981 to 0.1265) for lower color anomalies and stabilizes average motion intensity $A_{m}$ (Equation (Equation 2)) by reducing noisy high-amplitude frames (from 0.3758 to 0.0674 on sample subsets), enhancing feature consistency as evidenced by SRC gains in Table 5.

## 6. Conclusions

Our long-term action quality assessment (AQA) framework advances trampoline-gymnastics evaluation by integrating causal cross-modal attention with fine-grained temporal modeling. Using the Trampoline-AQA dataset, which includes synchronized RGB-optical flow data and expert annotations, we demonstrate that causal modeling can explicitly capture the temporal dependencies between 30 s action sequences, thereby revealing the propagation effects of early action errors on subsequent actions. We introduce the Temporal Feature Enhancer to enrich local motion details and the forward-looking causal cross-modal attention module to model dependencies among aerial maneuvers, establishing a robust foundation for intelligent assessment and autonomous coaching.

However, the framework faces scalability constraints due to the costly expert annotations required for Trampoline-AQA, as well as the computational expense of RAFT for optical-flow extraction, which, while accurate and mature, may limit real-time practicality. In future work, we will reduce annotation dependence via semi-supervised learning [61] and synthetic labeling pipelines, and enhance the time efficiency of the end-to-end system. Inspired by the later temporal attention mechanism proposed by Cai et al. [62], we plan to embed it into the FCCA module to emphasize dependencies in later action frames, enhancing the capture of error propagation in long sequences and improving prediction robustness. Continued architectural refinement and adaptive feature selection will further generalize the approach to complex movements, offering efficient and rigorous tools for coaching and assessment across diverse disciplines. 

## Figures and Tables

**Figure 1 sensors-25-05824-f001:**
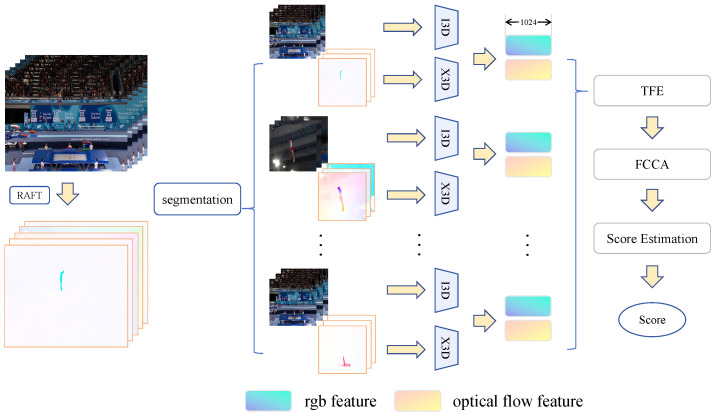
Overall pipeline of the proposed method. RGB frames are input to I3D, and optical flow is input to X3D for spatiotemporal feature extraction from each modality. The extracted features are then passed through the Temporal Feature Enhancer (TFE) to refine long-range temporal dependencies. Subsequently, the forward causal cross-modal attention (FCCA) module fuses the RGB and optical-flow features in a causally consistent manner, enabling robust score estimation for long-duration trampoline actions.

**Figure 2 sensors-25-05824-f002:**
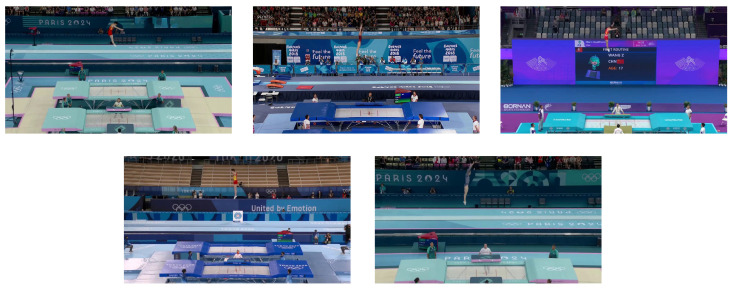
Competition scenes in the Trampoline-AQA dataset. The first row (**left** to **right**): 2024 Olympics Men’s Final, 2018 Buenos Aires Youth Olympics, 2022 Hangzhou Asian Games. The second row (**left** to **right**): 2020 Olympics Men’s Final, 2024 Olympics Women’s Qualifications.

**Figure 3 sensors-25-05824-f003:**
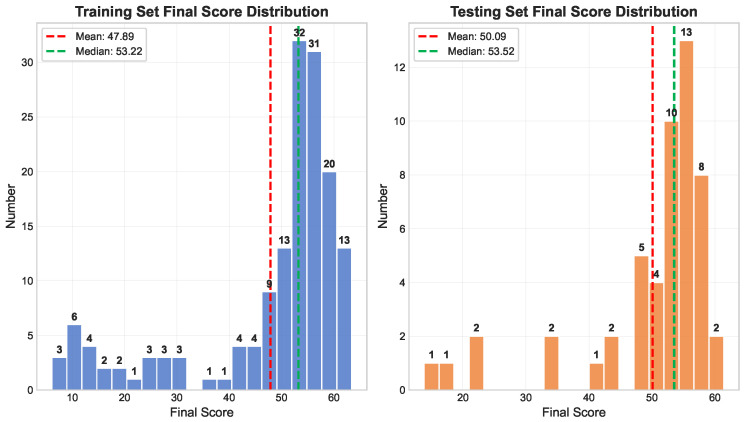
Statistical analysis of the Trampoline-AQA dataset, divided into training and test sets. The left figure shows the training set, and the right figure shows the test set. The red line indicates the mean score, and the green line indicates the median score.

**Figure 4 sensors-25-05824-f004:**
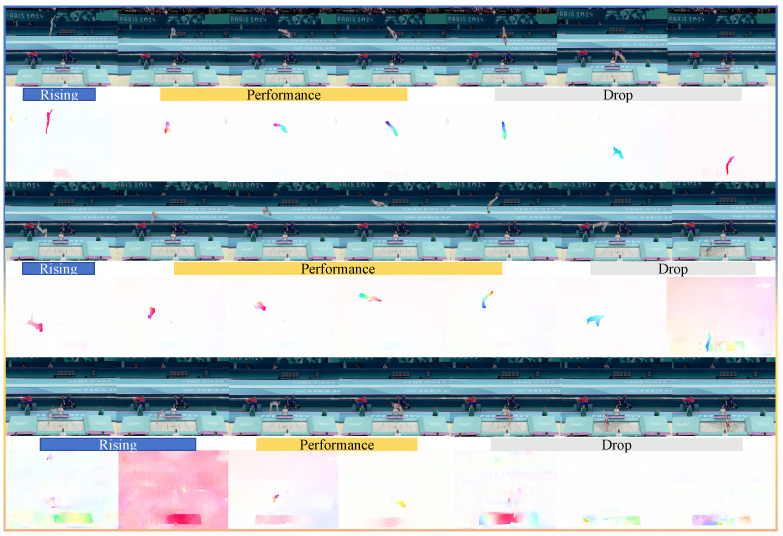
Joint analysis of video stream and optical flow for the Trampoline-AQA dataset. This figure shows the extracted visual features across three sequential phases: rising, performance, and drop. Each row represents a different data modality—the first corresponds to RGB frames, and the second to their optical-flow features. The first two rows depict normal feature-extraction results, capturing the raw visual appearance and motion dynamics, while the last row highlights abnormal cases in which feature-capture failures lead to distorted or incomplete representations. Specifically, normal cases exhibit natural and clear colors in video streams with continuous, smooth motion trajectories in optical flow. In contrast, abnormal cases show color distortion, overexposure, and blur in video streams due to high-speed motion, resulting in discontinuous vector distributions or blank areas in optical-flow maps.

**Figure 5 sensors-25-05824-f005:**
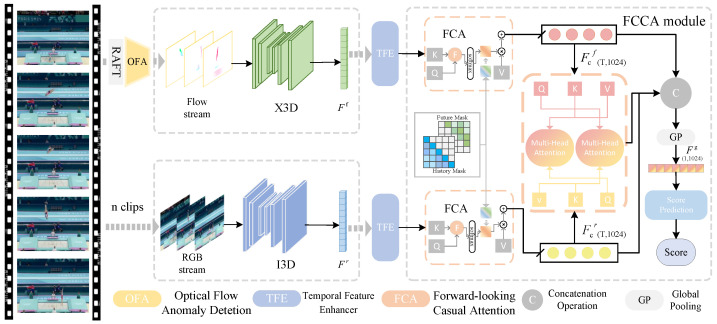
Architecture overview. The video stream is passed through I3D to extract features. The optical flow is extracted from the video using RAFT and the optical flow anomaly detection module (OFA), and X3D extracts features. The dual-stream features are first processed by the Temporal Feature Enhancer (TFE) to capture long-range temporal dynamics. Then, the forward-looking causal cross-modal attention (FCCA) performs causal and cross-modal fusion on the enhanced RGB and optical flow representations to generate discriminative embeddings. The features are then fed into the score autoencoder through global pooling to output scores.

**Figure 6 sensors-25-05824-f006:**
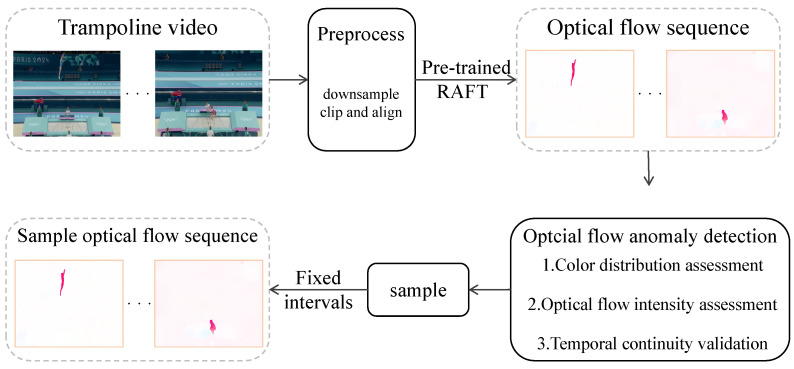
Flow chart of optical flow sequence extraction and sampling.

**Figure 7 sensors-25-05824-f007:**
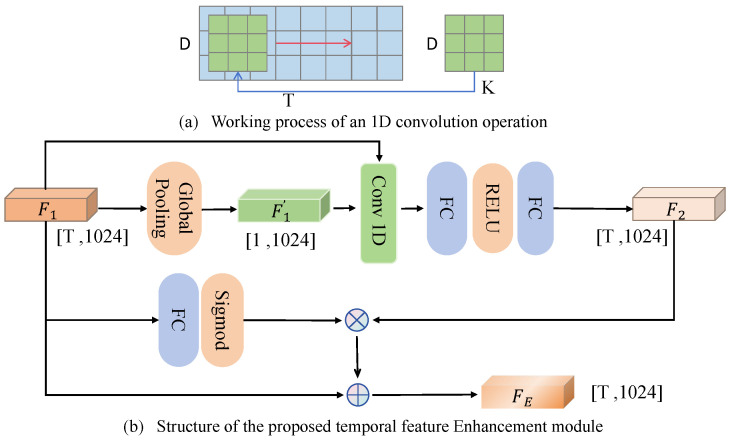
The proposed feature enhancement module; (**a**) illustrates the 1D convolution operation, where the kernel moves along the temporal dimension *T* with receptive field size *K* and depth *D*; (**b**) shows the structure of the temporal feature enhancement module, which processes the input feature $F_{1}\in R^{T\times 1024}$ to produce the enhanced feature $F_{E}\in R^{T\times 1024}$.

**Figure 8 sensors-25-05824-f008:**
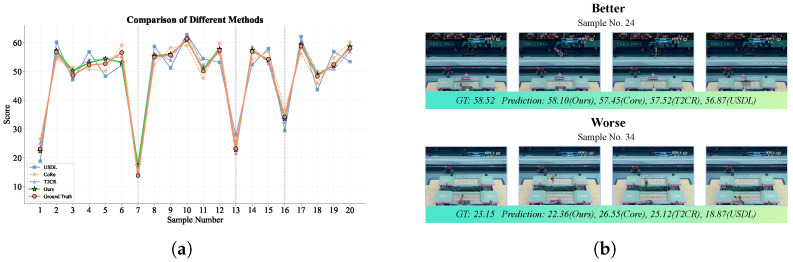
Visual analysis of the comparison of different methods. (**a**) Prediction comparison of our method and the competitors on the first 20 test samples. The relatively low scores on samples 1, 7, 13, and 16 correspond to visible performance errors (e.g., missteps, loss of balance) executed by the athletes, whereas the remaining samples contain fewer or no such errors. (**b**) Examples of predictions with the proposed method and other methods.

**Figure 9 sensors-25-05824-f009:**
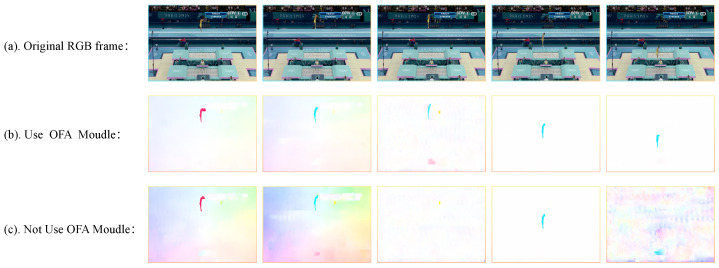
Visual analysis of optical flow processed by the OFA module. (**a**) Row: Original sampled video frame sequence. (**b**) Row: Optical flow sequence processed by the OFA module. (**c**) Row: Optical flow sequence without processing by the OFA module.

**Table 1 sensors-25-05824-t001:** Comparison between Trampoline-AQA and other sport-related AQA Datasets. “Avg. Duration” refers to the average duration of each video in seconds. “Labels” include annotations such as score, difficulty, action count, etc. “Samples” refers to the number of video instances, and “Year range” refers to the year range of the videos.

Dataset	Modality		Avg. Duration	Labels	Gender	Samples	Year-Range
	RGB	FLOW					
MIT Dive [13]	✓	×	6.0 s	Score	Male	159	2012
UNLV Dive [10]	✓	×	3.8 s	Score	Male	370	2012
UNLV Vault [10]	✓	×	3.8 s	Score	Male, Female	176	2012–15
AQA-7-Dive [7]	✓	×	4.1 s	Score	Male	549	-
MTL-AQA [8]	✓	×	4.1 s	Action, Score, Caption	Male, Female	1412	2012–14
FSD-10 [14]	✓	×	3–30 s	Action, Score	Male	1484	2020
FineDiving [9]	✓	×	4.2 s	Action, Score	Male	3000	2022
AGF-Olympics [11]	✓	×	-	Score, Difficulty, Gender	Male, Female	500	2008–20
Trampoline-AQA (Ours)	✓	✓	30.05 s	Score, Difficulty, Action Count	Male, Female	206	2018–24

**Table 2 sensors-25-05824-t002:** Trampoline-gymnastics competition tags and detailed annotations.

Label Type	Label Annotation
Competition Label	Competition type, Competition stage, Competition rank
Score Label	Difficulty, Execution, Flight time, Horizontal displacement, Penalty score, Total score
Other Label	Action count, Name, Nation, Gender, Classification label, Video duration

**Table 3 sensors-25-05824-t003:** Performance comparison on the Trampoline-AQA dataset.

Method	Feature	Modal	Year	SRC ↑ ^1^
C3D+LSTM [7]	I3D	RGB	2018	0.583
I3D+LSTM [31]	I3D	RGB	2019	0.551
USDL [18]	I3D	RGB	2020	0.759
CoRe [20]	I3D	RGB	2021	0.836
GDLT [50]	VST	RGB	2022	0.796
TSA [9]	I3D	RGB	2022	0.861
DAE [19]	I3D	RGB	2023	0.728
HGCN [21]	I3D	RGB	2023	0.699
T^2^CR [22]	I3D	Dual-stream RGB	2024	0.886
CoFInAl [41]	VST	RGB	2024	0.818
QTD [60]	VST	RGB	2024	0.829
FineParser [42]	I3D	RGB, Flow	2024	0.844
FineCausal [28]	I3D	RGB, Flow	2025	0.911
Ours	I3D	RGB, Flow	2025	0.938

^1^ SRC values range from −1 to 1 (higher is better).

**Table 4 sensors-25-05824-t004:** Performance comparison on the UNLV-Dive dataset.

Method	Modal	With DL SRC ^1^	Without DL SRC ^1^
C3D+SVR [10]	RGB	0.74	-
C3D+CNN [51]	RGB	0.80	-
S3D [53]	RGB	-	0.86
Adaptive [57]	RGB	-	0.83
FALCONS [55]	RGB	0.85	0.84
ScoringNet [52]	RGB	0.84	-
USDL [18]	RGB	-	0.81
MRSM (ESL) [58]	RGB	0.879	-
SCN+ATCN [56]	RGB	-	0.85
ESL+FTPSL [59]	RGB	0.87	-
OSL+FTPSL [59]	RGB	-	0.80
Ours (TFE+FCA)	RGB	0.882	0.863

^1^ DL: difficulty level; SRC values range from −1 to 1, higher is better. “With DL” refers to methods that accessed and utilized DL labels during training to optimize scoring objectives, but DL was not incorporated as a direct input feature into the model. “Without DL” refers to methods that do not use DL labels to optimize the scoring objective.

**Table 5 sensors-25-05824-t005:** Optical flow ablation experiment.

Method	Base Model	OFA	FCA	SRC ↑
Baseline	I3D			0.789
	X3D			0.806
Ours	I3D	✓		0.801
	I3D	✓	✓	0.853
	X3D	✓		0.821
	X3D	✓	✓	0.904

**Table 6 sensors-25-05824-t006:** RGB ablation experiment.

Method	Base Model	TA	FCA	SRC ↑
Baseline	I3D			0.759
	X3D			0.816
Ours	I3D	✓		0.822
	I3D		✓	0.837
	X3D	✓		0.862
	X3D		✓	0.893

**Table 7 sensors-25-05824-t007:** Temporal Feature Enhancer ablation experiment.

Modality	Base Model	TFE	SRC ↑
Optical flow	I3D		0.801
Optical flow	I3D	✓	0.882
Video stream	I3D		0.759
Video stream	I3D	✓	0.783

**Table 8 sensors-25-05824-t008:** Hyperparameter sensitivity analysis ablation experiment.

Temporal Kernel Size (k)	Attention Window (t)	SRC ↑
1	1	0.914
3	2	0.932
5	3	0.938
7	4	0.931

## Data Availability

The datasets used and/or analyzed during the current study are available from the corresponding author upon reasonable request. The video data used in this study were sourced from publicly available platforms. The dataset is permanently hosted on Zenodo https://doi.org/10.5281/zenodo.16195090 (accessed on 20 July 2025).

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
