# Peer review of "Enhancing Long-Term Action Quality Assessment: A Dual-Modality Dataset and Causal Cross-Modal Framework for Trampoline Gymnastics"

_sensors, 2025, doi:10.3390/s25185824_

Round 1
Reviewer 1 Report
Comments and Suggestions for Authors
In this work authors present a novel dataset Trampoline-AQA consisting of 206 video clips with two different modes namely, RGB and optical flow. This is done to address the existing inadequate database in modelling long duration, complex spatiotemporal sequence of actions in games like Trampoline. Authors present a framework for improving action quality assessment which comprises of blocks like Temporal Feature Enhancer (TFE), Forward looking Cross-modal Causal Attention and score auto encoder. Validation of framework is done on both in-house dataset and an external dataset. Authors report promising results for their proposed framework.
Strengths:
- Overall, the paper is well written, and the concepts are well explained before presenting the framework and experimental results.
- Long duration dataset (30.05 seconds) is key in efficient modelling of action dependencies.
- Dual modality (fusing RGB with optical flow) helped improve the quality assessment by overcoming noise.
- Causal Attention module (FCCA) took account of past actions and future inferences, improving the precision of action quality assessment.
- Ablation analysis established the credibility of each sub modules which are part of framework.
Scope for improvement
- Lines 8-9, what is the purpose of proposed framework? … not clear\
- In Line 77, 260, authors referred to backbones both I3D and X3D which is not included in in Figure 1, though included in Figure 5.
- Line 294, Eq. 2, I is not defined properly…is it Intensity function?
- In caption of Fig. 7, “…which processes the input feature F1 ∈ RT×1024 and processes it to produce the enhanced feature FE ∈ RT×1024….”…avoid redundancy for word “processes”
- In line 346, why X3D is not included?
- After line 363, in Eq. 4, term FC is used…not sure if it is defined earlier
- In line 398, what is “C Meanwhile” ?
- Av and Af in Eq. 9 and 10 need to be defined
- Figure 8a…authors mentioned executable mistakes…never elaborated on them before.
- Any reason for observation in line 507?
-
There is a lot of ongoing work in the last 3 years on improving Action Quality Assessment by considering spatio-temporal feaures as well as multimodal data. Authors elaborated these in Section 2.3 Though works addressed cross modality, none of them specifically focused on Trampoline game. Authors addressed this by validating their framework on custom built trampoline dataset. The only dataset which has longer samples (avg 2min) is AGF-Olympics…however they considered single modality.
-
Some Specific Concerns that need to be addressed by authors when handling cross modal attention
1) Is there a scope for token imbalance between RGB and Optical flow modalities? If no why ? If yes, how its affect/surpression on attention can be handled ?
2) Cross modality alignment issues are a major issue. How the proposed method addresses this issue ?
Reviewer 2 Report
Comments and Suggestions for Authors
Contribution: The paper introduces a new dual-modality dataset called Trampoline-AQA for long-duration action quality assessment in trampoline gymnastics. It proposes a novel causal cross-modal framework that includes a Temporal Feature Enhancer (TFE) and a Forward-looking Causal Cross-modal Attention (FCCA) module. The method significantly improves performance over previous baselines on both Trampoline-AQA and UNLV-Dive datasets.
This paper is of good quality, but needs to address the following comments:
- The authors mention the use of TFE and FCCA, but it is unclear how sensitive the model is to changes in the temporal kernel size and attention window. Could the authors share any hyperparameter sensitivity analysis?
- The optical flow anomaly detection module (OFA) plays a major role in preprocessing, but there is no runtime or computational overhead reported. How efficient is OFA in real-time applications?
- The paper introduces causal attention but does not detail how long-range dependencies are limited by the masking strategy. Could future masks bias score prediction based on early events?
- The dataset is impressive, but more clarity is needed on how subjectivity in human scoring was normalized across competitions and judges.
- The feature fusion strategy combines outputs from I3D and X3D models. How are these features aligned dimensionally and temporally before fusion? Any risk of modality imbalance?
- Suggest improving some of the figure captions — especially Figure 4 — to explicitly describe what makes a “normal” vs. “abnormal” case instead of relying on visual inference.
- Consider including error bars or statistical significance testing in Table 3 to support the claim of superiority over baselines.
- There are some writing issues, such as missing articles and awkward phrasing in the introduction and method sections. A final proofreading pass will help improve readability.
Reviewer 3 Report
Comments and Suggestions for Authors
1.This paper proposes a dual-modal dataset Trampoline-AQA for trampoline gymnastics, which provides a basis for follow-up research in field of paper research and related fields, and has great research significance.
2.This paper develops the TFE module to expand local temporal receptive fields through 1D convolutions, refining representations for complex sequences. The calculation of F'1 in formula (4) is inconsistent with the content in Figure 7. Please revise it.
3.This paper proposes the FCCA module to enhance spatiotemporal feature representation and improve the accuracy of long-term action quality assessment. Line 398, is the expression "C Meanwhile" a typo?
4.When performing feature extraction, this article selected two backbone models, I3D and X3D. It should be explained when to choose which model and what are the criteria for selection?
5.What is the baseline method used in sections 5.5.1 (including Table 5) and 5.5.2 (including Table 6)? It should be clearly stated.
Reviewer 4 Report
Comments and Suggestions for Authors
This paper proposed enhancing long-term action quality assessment for trampoline gymnastics. I think this paper can be accepted after revision.
- Please specify how judges’ scores were verified and the inter-rater reliability if available.
- The TFE module is inspired by [48], but the architectural details are sufficiently novel. Consider adding a brief comparison with [48] to highlight differences.
- The optical flow filtering visualization is effective. Could you quantify the improvement in optical flow quality after OFA (e.g., PSNR or SSIM)?
- While the dataset is highly valuable, the sample size (N=206) is relatively small compared to some other AQA datasets (e.g., MTL-AQA: 1412, FineDiving: 3000). The authors correctly split the data (75/25) and achieved excellent results, but the potential for overfitting or limited diversity remains a minor concern.
- In Table 4, the comparison is excellent. However, for full transparency, it should be clearly stated in the caption or a footnote whether the baselines "With DL" (Difficulty Level) used the DL label as an input feature to their model or merely had access to it during training. The superior performance of your method without explicit DL labels is a strong point that should be perfectly clear.
- How can later temporal attention inspire your future work? Later temporal attention, https://doi.org/10.1109/JBHI.2024.3384333, can be embedded into your methods..
- Discussion of Comparison to State-of-the-art action recognition methods including Deep historical long short-term memory network is missing. Add it.
- Using RAFT for optical flow extraction, while accurate, is computationally expensive. This impacts the practicality and scalability of the method.
Round 2
Reviewer 4 Report
Comments and Suggestions for Authors
The authors have addressed my concerns, and this paper can be accepted.